# Determinants of Quality of Life in Adult Patients with Chronic Non-Bacterial Osteomyelitis (CNO) of the Sternocostoclavicular Region (SCCH): A Dutch Single Center Study

**DOI:** 10.3390/jcm11071852

**Published:** 2022-03-27

**Authors:** Ashna I. E. Ramautar, Cornelie D. Andela, Neveen A. T. Hamdy, Elizabeth M. Winter, Natasha M. Appelman-Dijkstra

**Affiliations:** 1Division of Endocrinology, Department of Internal Medicine, Leiden University Medical Center, Albinusdreef 2, 2333 ZA Leiden, The Netherlands; c.d.andela@lumc.nl (C.D.A.); n.a.t.hamdy@lumc.nl (N.A.T.H.); e.m.winter@lumc.nl (E.M.W.); n.m.appelman-dijkstra@lumc.nl (N.M.A.-D.); 2Centre for Bone Quality, Leiden University Medical Center, Albinusdreef 2, 2333 ZA Leiden, The Netherlands

**Keywords:** chronic non-bacterial osteomyelitis, sternocostoclavicular hyperostosis, quality of life, SAPHO syndrome, patient-reported outcomes

## Abstract

Sternocostoclavicular hyperostosis (SCCH), the main clinical manifestation of chronic non-bacterial osteomyelitis (CNO) in adults, is associated with various degrees of chronic pain and restricted shoulder girdle function. We evaluated the impact of CNO/SCCH on quality of life (QoL) and its determinants in 136 adult patients with this rare auto-inflammatory bone disorder using the Short Form 36, Brief Pain Inventory, Brief Illness Perception, Utrecht Coping List, and Shoulder Rating questionnaires. Data were compared with those of the general Dutch population, patients with chronic pain, fibrous dysplasia, or osteoarthritis. Eighty-six (64%) predominantly female (85%) patients with completed questionnaires were included in the study. Sixty-four (75%) had isolated CNO/SCCH. Mean delay in diagnosis was 3.0 ± 5.5 (SD) years, 90% had variable pain, and 84% limited shoulder function. Compared to healthy and chronically diseased reference populations, CNO/SCCH patients demonstrated significant impairments in almost all aspects of QoL, maladaptive illness perceptions, and ineffective coping strategies. For patients with >5-year delay in diagnosis, higher pain scores and limited shoulder function were identified as determinants for impaired QoL. Patients with CNO/SCCH reported significant impairments in QoL associated with clinical and psychological determinants. Clinical measures such as shortening delay in diagnosis, effective pain management, and psychosocial interventions targeting these factors should help minimize the negative impact of CNO/SCCH on QoL.

## 1. Introduction

Chronic non-bacterial osteomyelitis (CNO) is a rare auto-inflammatory bone disorder of unknown origin. In adults, the lesions affect the axial skeleton, with a predilection for the medial end of the clavicles, sternum, and first ribs, which led to coining the term sternocostoclavicular hyperostosis (SCCH) to the disorder. Other areas of the axial skeleton such as the spine, sacrum, and mandible may also be affected. The skeletal lesions are associated with the characteristic dermatosis palmoplantar pustulosis (PPP) in about 30% of cases. The disorder most commonly presents in young adulthood, predominantly in women, with lesions most commonly isolated to the sternocostoclavicular region [1]. The natural history of CNO/SCCH is characterized by episodes of remissions and exacerbations, followed by a chronic state associated with potentially debilitating symptoms due to irreversible structural changes in affected bones, secondary arthritis in adjacent joints, and limitations in shoulder girdle function. These physical sequelae are often associated with psychological sequelae, which may both significantly impact various aspects of a patient’s quality of life [2].

Quality of life (QoL) is defined as “the functional effect of an illness and its consequent therapy upon a patient, as perceived by the individual patient” [3], and should thus always be determined from the perspective of the patient. A model used for describing patients’ understanding of their illness and the processes involved in behaviors for managing an illness’s threats, is the common-sense model of self-regulation (CSM), which provides a framework for examining cognitive and emotional representations of illness and health [4,5]. According to the CSM, QoL is influenced by two major determinants: illness perceptions, which consists of five cognitive components, and; direct influences on coping strategies, which are the skills used in handling different situations, crises, and life events that include emotional, behavioral, and cognitive responses [6,7,8,9,10,11].

A decade ago, our group conducted a study addressing QoL in a cohort of 52 adult CNO/SCCH patients, using structured interviews and validated QoL questionnaires, with findings demonstrating significant impairments in QoL compared to healthy controls. Our data identified a delay in diagnosis to be the major determinant of impairment in QoL due to its association with a higher prevalence of physical and psychosocial sequelae, such as higher emotional distress expressed as illness-induced anger, fear, distress, and dejection, and with more health-related impairment in social activities and to perceptions of more negative disease consequences [12].

However, available data on QoL remains scarce in CNO/SCCH, and the primary aim of our study was to further evaluate in depth various aspects of QoL and its determinants in our now larger cohort of adult CNO/SCCH patients and to compare these data with data from healthy and diseased reference populations. The second aim of our study was to evaluate the impact of identified clinical and psychological determinants in addition to diagnostic delays, such as pain severity, limitation in shoulder function, illness perceptions, and coping strategies on QoL in CNO/SCCH patients, as we hypothesized these determinants may also significantly impact on their QoL.

## 2. Material and Methods

### 2.1. Material

One hundred and thirty-six adult patients with an established diagnosis of CNO/SCCH based on characteristic clinical and radiographic features, who attended the outpatient clinic of the Leiden University Medical Center between 2017–2020, were invited to participate in this cross-sectional study by letters posted to their registered home address, including twelve who had participated in the study we conducted a decade previously [12]. The only inclusion criterion was the completion of >90% of five questionnaires (Figure 1).

### 2.2. Methods

Patients could complete the questionnaires online or by hand, returning them by post. Online data entry and control were performed through an online survey platform (NETQ, NETQ Healthcare B.V., Utrecht, The Netherlands). Up to two phone calls were placed to contact patients who failed to respond to the invitation sent by regular land mail.

Sociodemographic, clinical, and radiographic data were retrieved from the electronic medical records. Data on delay in diagnosis, follow-up years and level of education, localization and extent of skeletal lesions, and presence of extraskeletal manifestations such as palmoplantar pustulosis was also specifically collected. Level of education was categorized using the International Standard Classification of Education (ISCED), with a low level of education defined as a primary to lower secondary education; medium level of education as an upper secondary to post-secondary non-tertiary education; and high level of education as the first and second stage of tertiary education.

## 3. Questionnaires

### 3.1. Quality of Life

The Short Form 36 (SF-36) questionnaire is a validated widely used instrument for the evaluation of various domains of QoL in individuals older than 14 years. SF-36 consists of nine domains: “physical function”, “role physical function”, “bodily pain”, ”general health” perceptions, “vitality”, “social function”, “role emotional”, and “mental health”. Higher scores indicate better QoL [13].

The Brief Pain Inventory (BPI) questionnaire is an assessment tool originally designed to assess pain in patients with cancer [14] but also validated for the assessment of non-oncological pain, which has been widely used in clinical trials to evaluate pain due to different pathologies [15,16,17]. BPI domain scores include “pain severity” relating to pain intensity and “pain interference” relating to the impact of pain on functioning, which were both used as outcome parameters in our study. The response was measured using a rating scale from 0 to 10, with 0 = no pain or no interference and 10 = pain at its worst or completely interfering with physical, and/or emotional or social functions. BPI also includes a question about medication use and pain relief with the use of analgesics.

### 3.2. Illness Perceptions

The Brief Illness Perception questionnaire (BIPQ) was designed to evaluate the cognitive and emotional perception of illness in patients with chronic or acute conditions [18] and consists of nine items, each measuring a different illness. The first 5 dimensions assess cognitive illness representations such as consequences: effect on life; timeline—duration of illness; personal control—control over illness; treatment control—beliefs about the effectiveness of treatment, and; identity—experience of symptoms. Items 6 and 8 assess emotional representations including illness concern and a multifaceted question about emotional response and mood. Item 7 assesses coherence—degree of understanding of the illness. The first eight questions were rated using a 0 to 10 response scale, with higher scores indicating more negative or positive illness perceptions, depending on the item assessed. The final item 9 assesses causal representation using an open-ended response, asking respondents to rank the three most important causal factors for their illness.

### 3.3. Coping Strategies

The Utrecht Coping List (UCL) is a validated questionnaire widely used to evaluate coping strategies, based on the assumption that individuals prefer a certain way of coping, regardless of the situation—the coping strategy [19]. The UCL consists of 47 statements covering seven domains, each representing a different coping strategy. Each statement is scored on a four-point scale ranging from 1 (seldom or never) to 4 (very often). The seven domains have a different number of statements, so individual subscales have different maximum scores. Active coping (seven items, range 7–28) assesses a person’s ability to become aware of different points of view in addressing the problem, while confidently intending to solve it. Palliative coping (eight items, range 8–32) refers to not having to deal with a problem by seeking distraction with other occupations, such as smoking or drinking alcohol. Avoidant coping (eight items, range 8–32) entails not facing the problem and preferably attending to an issue as little as possible. Seeking social support (six items, range 6–24) assesses a person’s tendency to ask for help or comfort and understanding from family and/or friends. Passive coping (seven items, range 7–28) covers having a negative attitude towards the problem, feeling overwhelmed by it, or worrying about past experiences. Expressing emotions (three items, range 3–12), refers to someone’s tendency to show emotions such as anger or fear. Fostering reassuring thoughts (five items, range 5–20) refers to holding on to a positive attitude towards the problem, believing there are worse things in life. Higher scores indicate more negative or positive coping strategies depending on the item assessed [20].

### 3.4. Shoulder Function

The Shoulder Rating Questionnaire (SRQ) is a validated questionnaire used in the assessment of the severity of shoulder symptoms and degree of impairment of shoulder function [21,22,23]. The SRQ comprises 21 items measured in domains: pain, daily activities, recreational or athletic activities, work and satisfaction, and areas for improvement. Lower scores indicate less favorable outcomes.

### 3.5. Reference Groups

CNO/SCCH SF-36 data were compared with reference data from the general Dutch population (*n* = 1742) [24]; from a cohort of patients with osteoarthritis (*n* = 162) [25], and with data from our Center’s cohort of 97 patients with Fibrous Dysplasia/McCune Albright Syndrome (FD/MAS) [11,26]. CNO/SCCH SF-36 data were also compared between patients with localized (isolated) and more extensive skeletal involvement (additional axial localizations) and between CNO/SCCH patients and FD/MAS patients (isolated SCCH vs. monostotic FD and additional axial localizations vs. polyostotic FD). SF-36 data were also compared between patients with a delay in diagnosis of <3 years, 3–5 years, and >5 years.

BPI scores were compared between patients with isolated vs. additional axial localizations and between patients with shorter and longer delays in diagnosis. The impact of two BPI pain index scores, pain severity, and pain interference on QoL domains, was also evaluated. BIPQ data were compared with reference data from patients with two chronic disorders: asthma and diabetes mellitus type 2 [18].

CNO/SCCH coping strategies data were compared with those of two reference populations: a randomly selected group of Dutch women combined with a group of Dutch female nurses (*n* = 107, with median age 55 years (range 45–65)) [20], and a group of patients who experienced chronic pain of unknown cause, localized in the hip and/or knee reported to have occurred on at least three occasions over the previous month (*n* = 59, median age 64 years (range 55–74)) [27].

### 3.6. Statistical Analysis

Statistical analysis was performed using SPSS for Windows, Version 23.0 (SPSS, Inc., Chicago, IL, USA) Results are presented as means (±SD) or as median (range) and categorical variables were summarized with frequency counts and percentages.

The primary analysis included the comparison of SF-36, BPI, and BIPQ’s CNO/SCCH data with reference data using a pooled sample T-test. Secondary analysis included comparison SF-36, BPI, and BIPQ data between CNO/SCCH and FD/MAS subgroups based on the extent of skeletal lesions, and between CNO/SCCH subgroups based on the delay in diagnosis, by also using an unpaired *T*-test or Mann-Whitney U test. The final analysis included correlation analyses between the SF-36, BPI, SRQ, BIPQ, and UCL domain scores using Pearson’s and Spearman’s correlations for normally and non-normally distributed variables, respectively. The level of significance was set at *p* < 0.01 for all tests in order to correct for multiple testing.

## 4. Results

### 4.1. Patients Characteristics

Questionnaires were >90% completed by 86 patients (64%), *n* = 59 (69%) by hand and *n* = 41 (31%) online. Respondents were predominantly women (*n* = 73, 85%). The median age was 52 years (range 23–79 years), the median age at diagnosis was 43 years (range 22–69) years. Mean delay in diagnosis was 3.0 ± 5.5 years, and the median duration of follow-up after diagnosis was 5 years (range 0–26 years). A diagnostic delay of <3 years was observed in 36 patients (41%), 3–5 years in 23 patients (26%), and >5 years in 29 patients (33%). The level of education was ’low’ in 19 patients (20%), ‘medium’ in 37 patients (42%), and ‘high’ in 29 patients (33%). Patients had predominantly isolated CNO/SCCH (*n* = 64, 75%). Twenty-two patients (25%) had additional affected sites of the axial skeleton such as spine (*n* = 12, 41%), mandible (*n* = 7, 8%), or both (*n* = 3, 3%). Palmoplantar pustulosis was present in 27 patients (30%). Delay in diagnosis was shorter in patients with additional axial localizations compared to those with isolated CNO/SCCH (mean 3.5 ± 2.4 years) vs. 5.5 ± 6.2 years) (Table 1).

Twenty-six of the 56 working patients (46%) were unable to perform their work adequately and 17(21%) stopped working because of shoulder complaints.

### 4.2. Outcomes and Determinants of Quality of Life

CNO/SCCH patients had significantly lower QoL outcome scores compared to healthy individuals from the general Dutch population in all SF-36 domains, except for “mental ealth” (73 vs. 77, *p* = 0.029). Other QoL domain scores were respectively: “physical function” (68 vs. 83, *p* < 0.001), “role physical” (46 vs. 77, *p* < 0.001), “bodily pain” (53 vs. 75, *p* < 0.001), “general health” (53 vs. 71, *p* < 0.001), “ vitality” (57 vs. 69, *p* < 0.001), “social functioning” (64 vs. 84, *p* < 0.001), and for “role emotional” (72 vs. 82, *p* = 0.007).

Compared with patients with FD/MAS, a rare bone disease also associated with chronic pain and functional limitations, CNO/SCCH patients had more significant impairment in the domains: “role physical” (*p* < 0.001), “bodily pain” (*p* < 0.001), “social functioning” (*p* < 0.001), and “role emotional” (*p* = 0.007). In contrast, CNO/SCCH patients had significantly better scores compared to patients with osteoarthritis in the domains “physical function” (68 vs. 52, *p* < 0.001), “role physical” (46 vs. 33, *p* < 0.01), “vitality” (57 vs. 45, *p* < 0.001), “social functioning” (64 vs. 47, *p* < 0.001), “role emotional” (72 vs. 38, *p* < 0.001), and “mental health” (73 vs. 52, *p* < 0.001) (Figure 2).

On subgroup analysis, CNO/SCCH patients demonstrated a significant difference between isolated and more extensive axial lesions in the domains “role physical” (*p* < 0.001) and “role emotional” (*p* = 0.002), with intriguingly lower QoL scores observed in patients with isolated SCCH compared to those with more extensive lesions. Isolated CNO/SCCH patients also had significantly lower outcomes in all QoL domains (*p* < 0.001), except again for the domain “mental health”, compared to patients with similarly localized lesions in monostotic FD. However, there was no significant difference in QoL between patients with polyostotic FD and CNO/SCCH patients with additional axial localisations. There was no significant difference in SF-36 outcome scores, between patients with or without extra-skeletal manifestations, except for lower “general health” scores in the latter (48.0 vs. 55.8, *p* < 0.01) (Figure 2).

Whereas there was no significant difference in QoL domains outcomes between patients with a delay in diagnosis <3 years and those with a delay of 3–5 years (*p* > 0.05), patients with a delay in diagnosis >5 years had significantly more impairment in QoL compared to patients with a delay in diagnosis <3 years in the domains “physical function” (58 vs. 74, *p* > 0.001), “role physical” (35 vs. 54, *p* < 0.01), “bodily pain” (46 vs. 58, *p* < 0.01), “general health” (47 vs. 56, *p* < 0.01), and “social function” (56 vs. 68, *p* < 0.01). Patients with a delay of >5 years had significantly worse outcomes in “physical function” (58 vs. 73, *p* < 0.01) and also more impairment in “general health” (47 vs. 59, *p* < 0.01) compared to patients with a delay of 3–5 years (Figure 2).

Subgroup analyses of BPI outcome scores showed that pain scores did not significantly differ between isolated and non-isolated CNO/SCCH, and between patients with a delay <3 years and 3–5 years, although patients with isolated CNO/SCCH and a longer delay in diagnosis had consistently worse scores than patients with additional axial localisations and a shorter delay in diagnosis, albeit non-statistically significant. Patients with a delay in diagnosis >5 years had significantly worse pain outcome scores in all domains (*p* < 0.01) compared to patients with a delay <3 years (*p* < 0.01) (Figure 3). There was no significant difference in pain scores between patients with or without extra-skeletal manifestations (*p* > 0.05).

### 4.3. Shoulder Girdle Function Scores

Thirty-two patients (36%) had severe to very severe limitations in shoulder function during daily personal and household activities, and 39 patients (47%) had the same limitations only during recreational or sporting activities (Table 2).

### 4.4. Relationship between BPI Scores and SF-36 Scores

Higher BPI pain severity scores were associated with more significant limitations in SF-36 domains “physical function”, “role physical function”, “general Health”, “vitality”, “social function”, and “mental health” (all *p* = 0.001). Higher pain interference scores were also associated with perceiving more limitations in the domains “physical function”, “role physical function”, “general health”, “vitality”, “social function”, “mental health” (all *p* < 0.001), and “role emotional” (*p* = 0.002). However, neither of the BPI pain domains “pain severity” and “pain interference” were significantly associated with impairment in the SF-36 “bodily pain” domain, although higher BPI severity and interference index scores did show a trend for poorer QoL outcome scores in this domain (Appendix A).

### 4.5. Correlation between Shoulder Function and SF-36 Scores

Lower SQR scores, reflecting limitation in shoulder function, were significantly associated with lower outcome scores in SF-36 domains “physical function”, “role physical function”, “vitality” (all *p* < 0.001), “social function”, and “mental health” (*p* < 0.01). Limited shoulder function was also significantly associated with higher scores in all BPI pain domains (*p* < 0.001) (Appendix A).

### 4.6. Illness Perceptions

Compared to a reference group of patients with the chronic disease asthma (*n* = 309), patients with CNO/SCCH perceived more consequences of their disease, experienced less personal and treatment control, and attributed more of their symptoms to their disease (all *p* < 0.001). Compared to a reference group of patients with another chronic disease diabetes mellitus type II (*n* = 119), patients with CNO/SCCH also experienced less personal and treatment control of their illness, attributed more of their symptoms to their disease, had less concern about their illness, and had a worse personal understanding of their disease (all *p* < 0.001) (Table 3).

Subgroup analysis of patients with isolated CNO/SCCH compared to those with additional axial localizations showed that patients with isolated CNO/SCCH experienced less personal control of their illness (*p* < 0.01). Although patients with extra-skeletal manifestations had generally less favorable outcomes, there was no significant difference in the subscales of BIPQ between patients with or without extra-skeletal manifestations. Patients with shorter delays in diagnosis had a better personal understanding of their disease compared to patients with a delay >3 years (5.1 vs. 6.5, *p* < 0.01).

### 4.7. Relationship between Illness Perceptions and SF-36 Scores

Experiencing more consequences of CNO/SCCH and having more concern and emotional and cognitive representations of the disease were associated with greater impairment in all QoL domains (all *p* < 0.001) except for “bodily pain” (*p* = 0.103). Attributing more symptoms to the disease was associated with perceiving more limitations in “physical function”, “role physical function” and “general health” (all *p* = 0.001), and “social functioning” (*p* = 0.007). Perception of more “personal control” was associated with better “mental health” (*p* < 0.01). Less illness concern was associated with better SF-36 scores in all domains except for “bodily pain” (Appendix A).

### 4.8. Coping Strategies

There was no significant difference in coping strategies between CNO/SCCH and FD/MAS patients or with individuals from the UCL reference population, except for CNO/SCCH patients who reported expressing less emotions than the latter reference group (5.6 (1.4) vs. 6.8 (3.6), *p* < 0.01). Compared to patients with chronic pain due to other pathologies, patients with CNO/SCCH reported using more active coping strategies (19.0 (3.5) vs. 16.4 (4.0), *p* < 0.001), seeking more distraction (18. (3.1) vs. 16.8 (4.0), *p* < 0.01) and social support (12.8 (3.3) vs. 13.5 (3.7), *p* < 0.001), with no significant differences in other coping strategies (Figure 4).

Subgroup analyses showed that there was also no significant difference in coping strategies between patients with isolated CNO/SCCH compared to those with more extensive axial involvement, or between patients with shorter or longer delays in diagnosis.

### 4.9. Relationship between Coping Strategies and BPI and SF-36 Scores

Active coping strategies were correlated with a higher BPI pain severity index (r = 0.35, *p* < 0.001) and with worse SF-36 “general health” scores (r = −0.316, *p* < 0.01) (Appendix A).

## 5. Discussion

To our knowledge, this is the first study fully evaluating various aspects of quality of life and its clinical and psychological determinants in patients with the rare autoinflammatory disorder CNO/SCCH. Our findings demonstrate that adult patients with CNO/SCCH have significant impairments in nearly all aspects of QoL compared to a reference general Dutch population. Our data identify delay in diagnosis of >5 years, chronic pain of moderate to high severity, negative or maladaptive illness perceptions, and less effective coping strategies as the major determinants of QoL in CNO/SCCH, further consolidating previously published findings from our group [12].

Experiencing pain of variable severity is one of the major clinical manifestations of CNO/SCCH, and data from our study demonstrate that pain has a significant impact on various aspects of QoL, as reflected by the significant correlation between BPI outcome scores, especially those of “pain severity” and “pain interference” domains and most SF-36 QoL domains, surprisingly except for the “bodily pain” domain. This surprising outcome was further supported by a similar significant correlation between the SRQ shoulder function scores and all BPI pain scores and SF-36 domain scores, again except for “bodily pain”. These findings are of relevance in clinical practice, as they suggest that using SF-36 “bodily pain” domain outcome scores may not be sufficiently reliable for the evaluation of the impact of pain on QoL in patients with CNO/SCCH. A more complete evaluation of pain using additional BPI and SRQ outcome scores is thus recommended in the clinic for a full picture of the potentially significant contribution of pain to impairment of QoL in CNO/SCCH patients.

Interestingly, our data show that the main significant determinant of pain in CNO/SCCH is a delay in diagnosis rather than the extent of the axial lesions. Our data also show that a delay in diagnosis of >5 years is associated with significant impairment in all physical, social function, and general health domains, suggesting a likely contributory role of this delay in the development of irreversible structural changes, potentially leading to secondary degenerative changes, and chronic debilitating limitation of shoulder function and pain symptoms both having a significant impact on various aspects of QoL.

Our data further show a significant relationship between nearly all SF-36 domain scores and a number of illness perceptions domains such as “consequences”, “illness concern”, “emotional control’, and “identity”, the modulation of which may be potentially achieved by individually-tailored psychological therapy, which may, in turn, minimize the negative impact of maladaptive illness perceptions on QoL.

This is to our knowledge, the first study evaluating coping strategies in a relatively large cohort of adult patients with CNO/SCCH. Although active coping strategies were frequently reported, these were associated with generally worse SF-36 “general health” domain scores, suggesting that active coping and more personal control and understanding may not significantly influence QoL in adult CNO/SCCH. Seeking social support was also more frequently reported by patients with CNO/SCCH than in those with chronic pain of unknown etiology, possibly indicating that establishment of a diagnosis of CNO/SCCH as the source of the chronic pain may be associated with a greater sense of social acceptance and greater tendency to ask family and friends for help. More palliative coping, defined as seeking distraction with other occupations, such as smoking or drinking alcohol, was also more prevalent in CNO/SCCH patients, which is of particular relevance as smoking is a known trigger for an auto-inflammatory reaction, which may exacerbate disease activity, thus potentially inducing higher levels of pain and more limitations in shoulder girdle function [1].

We put our findings in perspective by comparing the impact of the identified clinical and psychological determinants of QoL in CNO/SCCH to the impact of these factors in other pathologies with similar characteristics as CNO/SCCH, such as chronicity in reference populations with asthma and diabetes mellitus type II, chronic pain in osteoarthritis, or similar pathological subtypes in FD/MAS, with lesions affecting one (monostotic/localized to SCC region) or more skeletal sites (polyostotic/other axial lesions)).

Another reason to perform this comparison was that as yet, there are no disease-specific questionnaires for CNO/SCCH so it may be difficult to judge whether an observed outcome is related to the disease itself, or to one of its symptoms such as pain. This provided the rationale for the comparison of CNO/SCH data with those of another chronic osteoarticular disease associated with pain such as OA. Since OA mostly affects older patients, the observed differences may be age-related, which was the rationale for using another reference population of patients with a rare chronic bone disease also associated with pain, but affecting younger patients such as FD/MAS.

Compared to a reference population of patients with chronic pain due to osteoarthritis, SF-36 QoL outcome scores were significantly less impaired in a number of domains in CNO/SCCH patients despite the fact that an inflammatory component is clearly involved in the pathophysiology of both disorders. Osteoarthritis is, however, a primarily generalized process of progressive degradation of cartilage matrix, associated with inefficient attempts at repairing, whereas adult CNO/SCCH is a rare auto-inflammatory bone disorder of the axial skeleton, with an abnormal osteogenic response to inflammation, eventually leading to irreversible structural changes in adjacent joints.

In contrast, patients with CNO/SCCH fared significantly less favorably compared to patients with FD/MAS in the QoL domains: “Role physical”, “bodily pain”, “social function” and “role emotional”, suggesting less adequate emotional adaptation for their disorder, possibly because the diagnosis of CNO/SCCH is usually established in adulthood, whereas FD/MAS is most commonly established during childhood, especially its more severe polyostotic/McCune Albright types. In contrast to patients with FD/MAS, in whom impairment in QoL was more pronounced in the presence of higher skeletal burden (polyostotic vs. monostotic disease), patients with isolated CNO/SCCH intriguingly demonstrated significantly worse scores in the physical and emotional domains of QoL than those with additional axial localizations. This discrepancy in the impact of the extent of skeletal involvement on QoL between the two pathologies may be due to the significantly longer delay in diagnosis observed in patients with isolated CNO/SCCH, which may allow the development of irreversible changes in affected bones and secondarily adjacent joints, and also due to the documented impact of mineralization defects and increased skeletal burden on an increased risk for pain, deformities and fractures in FD/MAS [26,28].

Comparing illness perceptions between patients with CNO/SCCH and those with type 2 diabetes or asthma, showed that all three groups were equally aware of the chronic nature of their disease. However, CNO/SCCH patients perceived more consequences of their disease than asthma patients but did not differ from patients with diabetes. A possible explanation for this is that diabetes and CNO/SCCH are conditions that are continuously present, whereas asthma usually manifests itself in acute exacerbations spaced in time. CNO/SCCH patients also experienced less personal and treatment control and attributed more of their symptoms to their disease than patients with asthma or diabetes (all *p* < 0.001). This probably reflects the fact that there is as yet no standard treatment for CNO/SCCH, and that various applied treatments are still often partially or not effective in controlling symptoms or disease progression. Finally, CNO/SCCH patients also had a worse personal understanding of their disease, and had less concern about their illness than those with diabetes (all *p* < 0.001), but did not differ in those aspects from patients with asthma. This may be due to the fact that diabetes is a much better-understood disease, its treatment is well-established, and patients are well aware of its possible serious (long-term) consequences if left untreated.

This comparison analysis provides valuable information about different ways by which similar physical and psychological determinants may impact QoL in CNO/SCCH patients compared to reference populations with other chronic diseases.

Our study has strengths as well as limitations. Of its main strengths is the inclusion of a relatively large number of well-characterized patients across the clinical spectrum of the distinctive adult CNO/SCCH phenotype, the prerequisite availability of near-complete sets of data from five validated questionnaires covering several aspects of QoL and pain, shoulder limitations, illness perceptions, and coping strategies, and the opportunity to compare CNO/SCCH QoL data with data from healthy and diseased reference populations. Put together, these data allowed us to identify the main determinants of impaired QoL in adult CNO/SCCH patients.

The main limitations of our study are that all questionnaires used were not CNO/SCCH disease-specific, although validated generic and domain-specific questionnaires, and that SRQ outcomes were not validated by objective clinical evaluation of shoulder girdle function at the time of completion of the questionnaire.

## 6. Conclusions

In conclusion, findings from our cross-sectional study demonstrate that adult patients with CNO/SCCH have significant impairment in almost all aspects of QoL, maladaptive illness perceptions, and ineffective coping strategies, compared to healthy and chronically diseased reference populations. The main identified determinants of impaired QoL in adult CNO/SCCH were delay in diagnosis, chronic pain of moderate to high severity, and restricted shoulder girdle function. We believe findings from this study may hold significant clinical implications in the management of CNO/SCCH patients, such as shortening delay in diagnosis, effective pain management, early institution of adequate therapies to decrease disease activity, and appropriate psychological measures to address altered illness perceptions and inadequate coping strategies may help to minimize the negative impact of CNO/SCCH on various aspects of quality of life in patients with this rare chronic auto-inflammatory bone disorder.

## Figures and Tables

**Figure 1 jcm-11-01852-f001:**
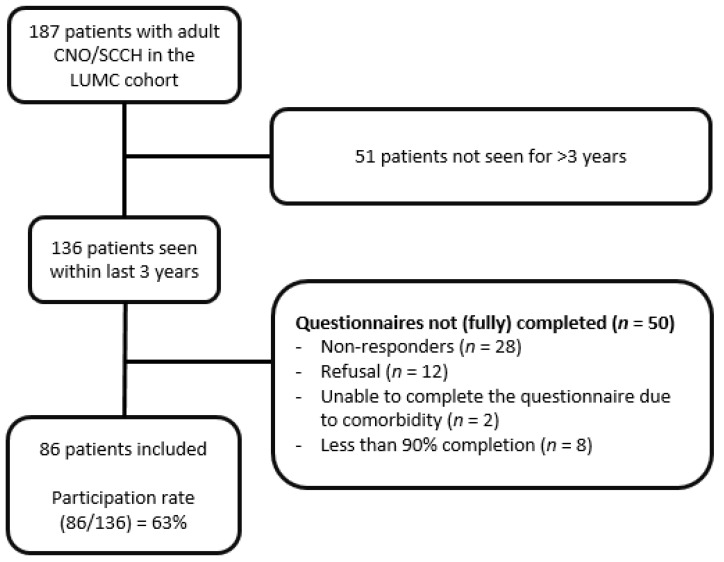
Flowchart of patient inclusion and participation rate. CNO/SCCH: Chronic Non-Bacterial Osteomyelitis/Sternocostoclavicular hyperostosis, LUMC: Leiden University Medical Center.

**Figure 2 jcm-11-01852-f002:**
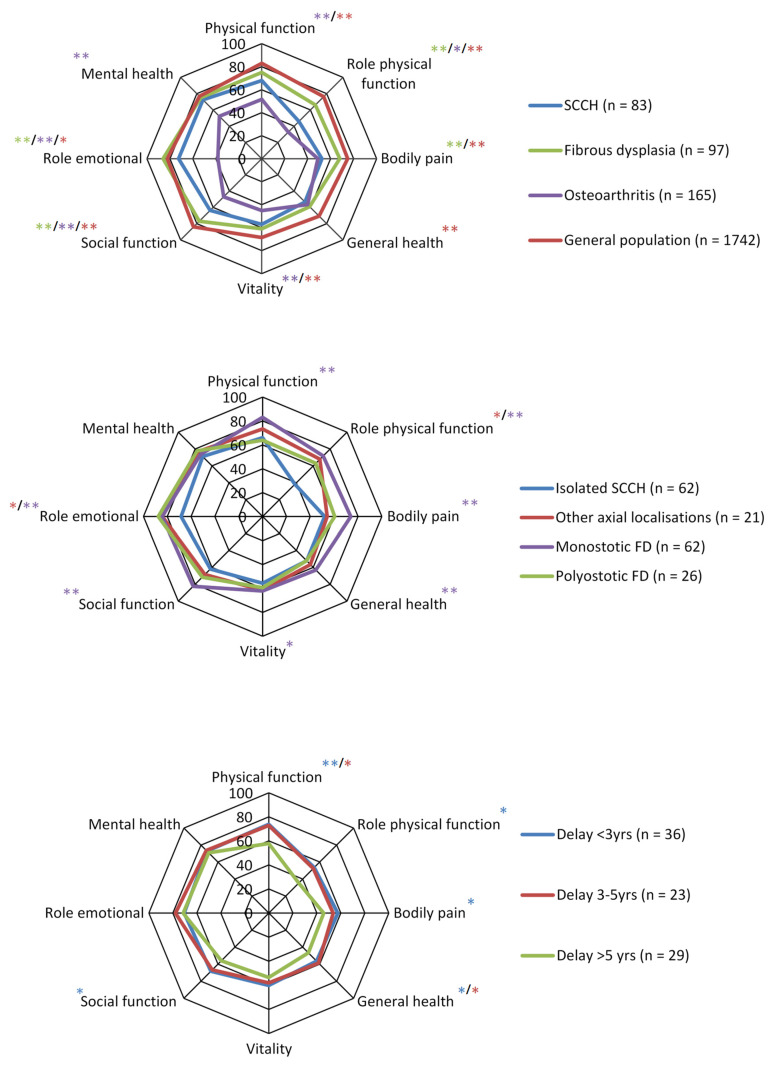
Radar charts comparing QoL Short Form-36 scores between CNO/SCCH patients and the general Dutch population, Fibrous Dysplasia (FD) and Osteoarthritis patients (OA), between subtypes of CNO/SCCH (isolated SCCH vs other axial localisations) and FD (isolated SCCH vs. monostotic FD), and between the length of delay in diagnosis (<3 yrs/3–5 yrs and >5 yrs) to a longer delay in diagnosis (>5 yrs). Significance was illustrated by * *p* < 0.01 or ** *p* < 0.001. The colors represent different subgroups in the different spider graphs.

**Figure 3 jcm-11-01852-f003:**
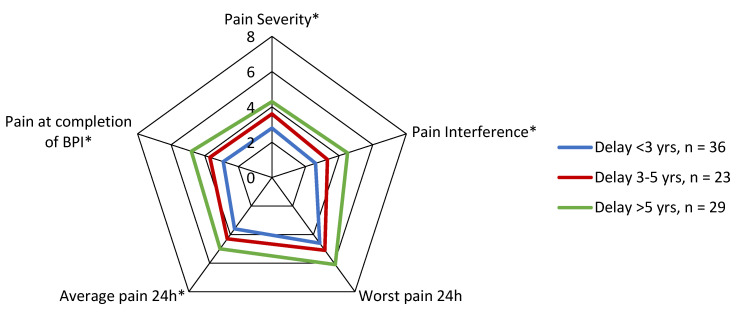
Radar chart comparing Brief Pain Inventory scores between CNO/SCCH patients with different lengths of delay in diagnosis <3 yrs and >5 yrs. Significance was illustrated by * *p* < 0.01.

**Figure 4 jcm-11-01852-f004:**
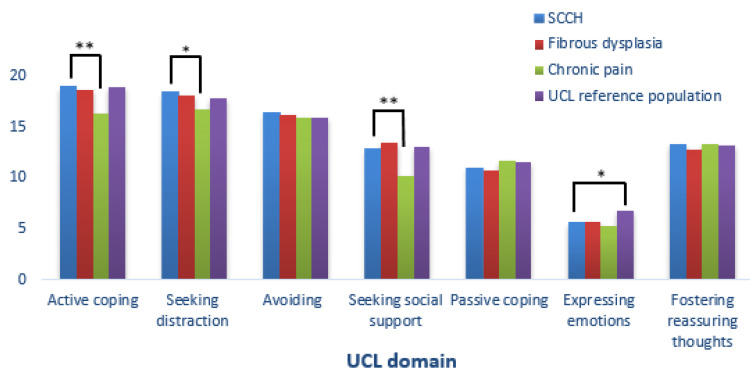
Bar chart comparing the UCL (Utrecht Coping List) scores between CNO/SCCH patients and the UCL reference population, Fibrous Dysplasia, and patients with chronic pain. Significance was illustrated by *****
*p*  <  0.01 or ******
*p*  <  0.001.

**Table 1 jcm-11-01852-t001:** Patient characteristics of adult patients with chronic non-bacterial osteomyelitis of the sternocostoclavicular region (CNO/SCCH).

	*n* = 86
Gender (male/female)	13/73 (15/85%)
Age (yrs) Age at diagnosis (yrs)	52 (range 23–79) 43 (range 22–69)
Delay in diagnosis (yrs)	3.0 (±5.5 SD)
<3	36 (41%)
3–5	23 (26%)
>5	29 (33%)
Educational Level	
Low	19 (20%)
Medium	37 (42%)
High	29 (33%)
Unknown	4 (5%)
Follow-up (yrs)	5 (range 0–26)
Subtypes of CNO/SCCH	
Isolated CNO/SCCH	64 (75%)
Additional axial localizations	22 (25%)
Spine	12 (14%)
Mandible	7 (8%)
Both	3 (3%)
Extraskeletal manifestations	27 (30%)

Data expressed as median (range), mean (SD), or number and percentage.

**Table 2 jcm-11-01852-t002:** Outcome of the Shoulder Rating Questionnaire scores in CNO/SCCH patients.

Limitation of Shoulder Function	During Daily Personal and Household Activities	During Recreational or Athletic Activities
(Very) severe limitation	32 (36.4)	39 (46.6)
Moderate	25 (28.4)	25 (28.4)
Mild	17 (19.3)	13 (14.8)
No	14 (15.9)	9 (10.2)

Data expressed as number and percentage (%) of patients.

**Table 3 jcm-11-01852-t003:** Comparison of Brief Illness Perception Questionnaire scores between SCCH patients and different patient groups.

B-IPQ	SCCH *n* = 86	Asthma *n* = 309	Diabetes 2 *n* = 119
Consequences	5.1 (2.8)	3.5 (2.3) ^α^	4.7 (2.9)
Timeline	8.9 (2.0)	8.8 (2.2)	9.2 (1.9)
Personal control	4.8 (2.7)	6.7 (2.4) ^α^	6.7 (2.3) ^α^
Treatment control	6.1 (2.5)	7.9 (2.0) ^α^	8.0 (2.3) ^α^
Identity	5.9 (2.6)	4.5 (2.3) ^α^	4.6 (2.8) ^α^
Illness concern	4.7 (2.7)	4.6 (2.8)	7.0 (3.1) ^α^
Coherence	5.8 (2.6)	6.5 (2.6)	7.9 (2.3) ^α^
Emotional response	4.2 (2.9)	3.3 (2.9)	4.3 (3.3)

Data are mean (SD). ^α^
*p*  <  0.001 compared to SCCH patients.

## Data Availability

Data supporting reported results can be provided by request to the corresponding author.

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
