# Peer review of "Determinants of Quality of Life in Adult Patients with Chronic Non-Bacterial Osteomyelitis (CNO) of the Sternocostoclavicular Region (SCCH): A Dutch Single Center Study"

_jcm, 2022, doi:10.3390/jcm11071852_

Round 1

Reviewer 1 Report

The article investigates the quality of life of patients with chronic non-bacterial osteomyelitis of the sternocostoclavicular region. The topic is certainly interesting and there is little evidence in the literature on the subject. The study was methodologically well conducted. The article is clear and well written. The conclusions are consistent with the results. I have no more suggestions for the authors

Author Response

See the uploaded file 'Response to reviewers'.

Reviewer 2 Report

This is a well written manuscript regarding the evaluation of the quality of life in the sternocostoclavicular hyperostosis due to chronic non-bacterial osteomyelitis.

Methods are adequately described. Results are well presented. 

Author Response

(The authors gave the same response as above.)

Reviewer 3 Report

The study is an interesting analysis regarding the impact of an inflammatory skeletal pathology on the patient's quality of life. Although the report provides important data through validated instruments, the manuscript presents some problems that must be solved, which are addressed below, item by item when necessary.

1. In the abstract, it may be necessary to define the acronym QoL. Although the meaning is evident in the main text, it does not fulfill its function in the summary text.
2. In the introduction, an adequate background is presented regarding the pathology and the definition of the study variable; additionally, the authors refer to their previous experience with a similar study. The presentation of the objectives of the work is adequate, and the scope as well.
3. The methodology adequately describes the instruments used to carry out the study, which allows us to understand their purpose. In addition, control or comparative groups seem adequate.
4. In the results, several tables are presented, and the graphs adequately show the data distribution. However, there are several details that the authors should improve:
- Table S1 is a bit confusing. The reader must guess in the second column what each data represents. This table should facilitate the overview of the data, so it is suggested to improve its presentation.
- Line 188 says "3.0   5.5", some information seems to be missing in the space between the numbers (this is also noted in the abstract)
- Line 191 says (32%) but in the table, the same information indicates 33%.
- In Table S3 is not possible to observe the complete information.
5. In the discussion, the authors broadly address the subject, which speaks of the good experience of the group.
6. The conclusion is appropriate to the expected results.

Congratulations to the authors for the work done.

Even with the above, I must note that this review was not double-blinded. The name of the authors is written in the article. I do not know if this is journal policy, but it is the first time it has happened to me.

Author Response

(The authors gave the same response as above.)

Reviewer 4 Report

This paper seeks to evaluate the impact on quality of life and its determinants
in 136 adult patients with Sternocostoclavicular hyperostosis. The article is well written. However, the second analysis they provide by comparing data with other pathologies such as OA, Asthma, Diabetes 2, fibrous dysplasia, chronic pain in different graphs and charts does not help the reading and blurry the message. In addition, the massive amount of correlation performed and of significance results found out does not help draw a take-home message. The paper will have been better with a clear hypothesis. 

Major comments:
Since you present a lot of correlations, please investigate them deeper. 
For instance, perform a Stepwise regression that uses statistical significance to select the explanatory variables in a multiple regression model.

Please place the graphs and tables related to the comparison with other data in the supplemental material and present the results only in a paragraph in the text.

What is the take-home message, and how will this be useful for clinicians? Please place your results in the clinical context. 

Minor comments: 

Some typo in tables and misalignment needs to be corrected, 
Please reorganize your discussion by focussing on the results you obtained through your questionnaire, then compare them with other pathologies. 

Author Response

see the uploaded file 'response to reviewers'.

We have addressed the (minor and major) comments raised by Reviewer 3 and 4 and have amended the manuscript accordingly.

More specifically as also addressed in more details in Response to Reviewer, we have addressed each of the points raised as follows:

  1. The take home messages are clearly outlined in Abstract and Conclusion and further addressed in the Discussion section of the manuscript.
  2. Tables and text typos and misalignments have been addressed.
  3. We have restructured the Discussion section according to the Reviewer’s suggestions, focusing first on findings in CNO/SCCH, then comparing findings in CNO/SCCH with those of other pathologies. Changes in the text are shown as track changes in the revised version of the manuscript.

We hope that these amendments satisfactorily address Reviewer 3/4’s comments.

Round 2

Reviewer 4 Report

The authors did not address all the comments of the reviewer. 

They are inconsistencies in the level of significance. Sometimes it's p < 0.01 and p < 0.05, there is no reason for this, which creates useless confusion.

Since the authors decided not to perform multiple linear regression analysis to identify which subsets of explanatory variables may contain redundant information about the response, the authors are making conclusions on unnecessary overlapping data.

They provide more than 75 significant correlations at p<0.01 ( why not 0.05 ?)
What is the conclusion based on this? If the authors cannot explain these results and how they can be helpful, this is just a list of useless data.

The massive amount of correlation presented does not bring any factual information and should be considered in the supplemental material.

Figure S4 is hard to read and just a copy and paste of bad quality excel sheet.

Regarding the comparison with other diseases, the authors insist on keeping them, but it does not bring anything in this paper. For instance, they discuss the comparison with OA and not with the other pathologies presented in the results; this does not make any sense.

If the authors want to compare their results to other pathologies, they could write another paper with that goal.

Author Response

Please find the Response to reviewer attached.
